# ECG Heartbeat Classification Using CONVXGB Model

**Atiaf A. Rawi [1,*], Murtada K. Elbashir [2]** and **Awadallah M. Ahmed [1]**

1   Department of Computer Sciences, Faculty of Mathematical and Computer Sciences, Gezira University, P.O. Box 20, Wad Madani 21111, Sudan; awadallah@uofg.edu.sd
2   Department of Information Systems, College of Computer and Information Sciences, Jouf University, Sakaka 72388, Saudi Arabia; mkelfaki@ju.edu.sa
*   Correspondence: atiaf.ayal88@gmail.com

**Abstract:** ELECTROCARDIOGRAM (ECG) signals are reliable in identifying and monitoring patients with various cardiac diseases and severe cardiovascular syndromes, including arrhythmia and myocardial infarction (MI). Thus, cardiologists use ECG signals in diagnosing cardiac diseases. Machine learning (ML) has also proven its usefulness in the medical field and in signal classification. However, current ML approaches rely on hand-crafted feature extraction methods or very complicated deep learning networks. This paper presents a novel method for feature extraction from ECG signals and ECG classification using a convolutional neural network (CNN) with eXtreme Gradient Boosting (XBoost), ConvXGB. This model was established by stacking two convolutional layers for automatic feature extraction from ECG signals, followed by XGBoost as the last layer, which is used for classification. This technique simplified ECG classification in comparison to other methods by minimizing the number of required parameters and eliminating the need for weight readjustment throughout the backpropagation phase. Furthermore, experiments on two famous ECG datasets–the Massachusetts Institute of Technology–Beth Israel Hospital (MIT-BIH) and Physikalisch-Technische Bundesanstalt (PTB) datasets–demonstrated that this technique handled the ECG signal classification issue better than either CNN or XGBoost alone. In addition, a comparison showed that this model outperformed state-of-the-art models, with scores of 0.9938, 0.9839, 0.9836, 0.9837, and 0.9911 for accuracy, precision, recall, F1-score, and specificity, respectively.

**Keywords:** ECG; CNN; XGBoost; ConvXGB; myocardial infarction (MI); arrhythmia; deep learning

## 1. Introduction

Every year, more than 17 million people die from cardiac diseases, which collectively remain the leading cause of mortality worldwide. According to the World Heart Federation, approximately 75% of all cardiovascular disease (CVD) patients reside in low-income communities [1]. Electrocardiography (ECG) is the most accurate and trustworthy tool available for diagnosing cardiac disease since it is non-invasive and accurately represents the electrical rhythm of depolarization and repolarization of the cardiovascular system. ECG captures the electrical activity created by depolarizing the heart muscle, propagating pulsing electrical waves towards the skin. Even though the energy level involved is low, it may be successfully detected using sensors connected to the chest [2]. The earlier an irregular cardiac rhythm is identified, the less severe the consequences are, and the faster the patient recovers from the condition [3]. However, ECG signals have complicated and highly chaotic properties, making their interpretation time-consuming and laborious, even for experienced professionals [4]. As a result, computer-assisted approaches are necessary to ease human workloads and eliminate misinterpretations caused by fatigue, differences between operators, and operator-specific mistakes, among other factors.

Machine learning (ML) is an essential tool for predicting and diagnosing deadly illnesses [5,6]. As a sub-branch of ML, deep learning (DL) has yielded outstanding results in the medical field, sometimes even outperforming physicians [7], because its hi-

erarchical structure allows substantial and high-level feature extractions that improve classification accuracy.

In tackling machine learning challenges, feature learning has emerged as a critical success factor [8]. However, most ECG classification methods depend on hand-crafted techniques for feature extraction using signal processing tools and methods such as filters [9], Fourier transforms, and wavelet transformation [10,11]. ML classifiers such as support vector machines (SVMs) have been utilized for classification [12]. The separation of these methods' feature extraction and pattern classification components is their main disadvantage. Additionally, these methods require domain experience with the processed data, and the attributes must be chosen. Furthermore, extracting features with the aid of experts takes time, and features may not be resistant to noise, resizing, or transformation, which means that they may not generalize well to new data [13]. As a result, we must recognize the significance of effective models and the capacity to acquire new features automatically in order to develop a comprehensive feature extraction and classification model. Most shallow and classical DL models rely on a single model for their initial training. Many researchers have recently shown an interest in the performance of deep neural networks in interpreting ECG signals, particularly convolutional neural networks (CNNs) that use one-dimensional (1D) and two-dimensional (2D) convolution to enhance their performance. DL models may learn invariant and hierarchical features directly from data, with ECG signals as input and class prediction as output. Recurrent neural networks (RNN), CNNs [14], and autoencoders are utilized for 1D ECG categorization. The input ECG data are converted into images or 2D representations for 2D ECG classification. Experiments have shown that the accuracy of 2D ECG classification is better than that of 1D ECG classification [15]. This paper presents a new method for classifying ECG signals, inspired by previous work [16], that takes advantage of two types of models and avoids their disadvantages, resulting in a better overall model. The proposed ConvXGB model is a new DL model for ECG classification that combines the performance of a CNN with XGBoost. As the results of this study demonstrate, ConvXGB performs better than either CNN or XGBoost alone and performs better than state-of-the-art models. The reasons for choosing these two model types were as follows:

1. XGBoost is a scalable ML approach for tree boosting designed to prevent the overfitting of data. It performs well on its own and in a variety of ML contexts. However, there is some uncertainty about the effectiveness of this method with respect to feature learning.
2. The use of CNN, a DL class with multiple levels of hierarchical learning, improves the clarity of the results.

This combination of methods has been tested on many datasets and has been shown to solve classification problems more accurately than other methods [16]. To overcome the shortcomings of existing methods for ECG signal classification, a new model, referred to as ConvXGB, is proposed that handles the feature extraction perfectly and reduces the number of parameters required to achieve the best performance results among the methods compared with the lowest processing time and effort. The most significant contributions of this paper can be summarized as follows:

1. We propose an end-to-end method for ECG signal classification (tested on two commonly used datasets) without the need for any complicated signal pre-processing.
2. The proposed method is suitable for deployment in low-computational-power devices (such as mobile phones) because many hyperparameters are needed to reduce the prediction time (0.6 ms).
3. We achieved better results with this hybrid method than the existing state-of-the-art method in terms of accuracy, precision, recall, F1-score, and specificity, and the lowest false-negative and false-positive rates.
4. To demonstrate the robustness and generalization ability of the proposed method, we tested it on a dataset from a different source and achieved highly accurate results.

The remainder of this paper is structured as follows: Section 2 provides a brief overview of studies relevant to ECG categorization. The methods used in this research are described in Section 3. Section 4 presents the experiment conducted. Section 5 discusses the results obtained. Finally, we present conclusions drawn from the result in Section 6.

## 2. Related Work

ML algorithms are used widely in ECG signal classification. ML classifiers such as SVMs have been shown to perform better in ECG signal classification than other algorithms, such as neural networks (NN), random forests (RF), and Bayesian algorithms [17,18]. In [19], an SVM was merged with linear discriminant analysis to produce a classification method for six arrhythmia types. Ref. [20] presented a multi-layer perceptron (MLP) model, which exhibited good performance. Ref. [21] presented a 1D CNN technique for five types of arrhythmia classification. A wavelet is first used for noise removal, and then a 1D CNN model is used for feature extraction. A fully connected layer with a softmax activation function is used for classification. The same 1D CNN method was used in [22] to classify four ECG signal types with denoising pre-processing steps. Current methods perform signal pre-processing, feature extraction, and prediction [23–27]. Various algorithms are used to perform these actions [28–33]. Other studies have used a deep belief network for ECG signal classification and thereby reduced false negatives effectively [34–37]. Stacked autoencoders have been used in other studies [38,39]. In [40,41], CNN and long short-term memory (LSTM) were used to build the encoding and decoding layers. Residual neural networks (RNNs) have been used to classify ECG signals and handle time-series data. RNN models have also been combined with other DL models, and LSTM has been combined with CNN [42]. Additionally, bidirectional LSTM has proven to be successful in classifying ECG signals because of its ability to process data in both the previous and forward directions [43].

It can be observed that most of the earlier mentioned works are based on machine learning methods after feature extraction steps that require domain knowledge and are time-consuming, or based on deep learning methods, which require a large amount of data and a long training time. Our suggested method uses both methods by using CNN for feature extraction (without training) and XGBoost classifier for classification. This technique reduced the training process time and achieved high results.

## 3. Materials and Methods

The overall classification process is illustrated in Figure 1, using the MIT-BIH dataset (in the upper part of Figure 1 for classifying five ECG signal classes) and the PTB dataset (in the lower part of Figure 1 for classifying two classes). For each dataset, the signals are fed into the hybrid ConvXGB model for classification.

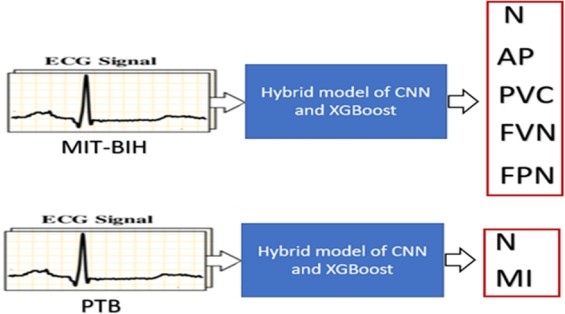

**Figure 1.** Illustration of the proposed method. The ECG signals fed into the suggested model are to be classified. The upper half of the figure shows the MIT-BIH dataset used for multiclass classification into five classes, while the lower half shows the PTB dataset for binary classification.

The proposed method has two main parts. The first part consists of three convolutional layers and a max pool layer for feature extraction, and the second part uses the XGBoost classifier for classification. Details of the architecture and model layers are shown in Figure 2.

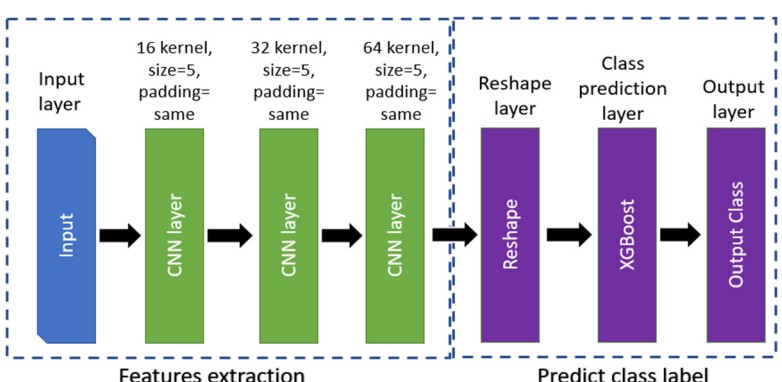

**Figure 2.** Architecture of the ConvXGB model.

### 3.1. CNN Model

As mentioned earlier, the convolutional neural network represents the first part of the proposed method. For input data $\mathfrak{D}$ in $r$ rows and $c$ columns, $\mathfrak{D} \in \mathbb{R}^{rXc}$ can be defined as shown in Equation (1):

$$\mathfrak{D} = \{x(i,j)|1 \leq i \leq r, i \leq c\}, \tag{1}$$

where $x(i,j)$ is the data value at $(i,j)$. When a kernel (or filter) $\varkappa$ of dimension $w_f X h_f$ is applied by a stride, $s_f$, the product can be defined as shown below:

$$(\mathfrak{D} \otimes \varkappa)_{i,j} = \sum_{u=-w_f}^{w_f} \sum_{v=-h_f}^{h_f} k_{u,v} D_{i+u+v}. \tag{2}$$

For each convolutional layer $l$, a bias will be added and a convolution operation applied. For a feature map indexed by $f \in \{1, \dots f(l)\}$, the output, $\mathbb{y}_n^l$, of the $l$th layer for the $n$th feature map is obtained from the output of the previous layer, $\mathbb{y}_n^l$, as follows:

$$\mathbb{y}_n^l = \Theta\left(B_n^l + \sum_{p=1}^{f(l-1)} k_{n.p}^l \times \mathbb{y}_p^{l-1}\right), \tag{3}$$

where $\Theta$ is the used activation function (Relu), $B_n^l$ is the bias matrix, and $k_{n.p}^l$ is the filter of size $2w_f + 1 \, X \, 2h_f + 1$. The output of layer $l$ for the $n$ feature map, $\mathbb{y}_n^l$, at position $(i,j)$ is therefore as follows:

$$\left(\mathbb{y}_n^l\right)_{i,j} = \Theta\left(\left(B_n^l\right)_{i,j} + \sum_{p=1}^{f(l-1)} (k_{n.p}^l \otimes \mathbb{y}_p^{l-1})_{i,j}\right) = \Theta\left(\left(B_n^l\right)_{i,j} + \sum_{p=1}^{f(l-1)} \sum_{u=-w_f^l}^{w_f^l} \sum_{v=-h_f^l}^{h_f^l} (k_{n.p}^l)_{i,j} (\mathbb{y}_p^{l-1})_{i+u,j+v}\right) \tag{4}$$

To avoid overfitting and downsampling the features extracted, a pooling layer is applied, which replaces the output with the average or maximum value of a sliding window. For example, by applying the pooling function $P(.)$, the output will be defined as shown below:

$$P(\mathbb{y}_n^l)_{i,j} = max(\mathbb{y}_n^l)_{i,j}, \tag{5}$$

The output is then classified using XGBoost in the next part of the method.

### 3.2. XGBoost Model

XGBoost is a machine learning method for classification and regression problems developed by Chen and Guestrin [1]. It is a massively efficient approach for classification and regression problems and uses a tree ensemble to improve performance. The ensemble summation of all $T$ and regression trees (CARTs), where each tree has $T_E^i \mid i \in 1 \dots T$ nodes, is as follows:

$$y_{hat_i} = \sigma(x_i) = \sum_{t-1}^{T} f_t(x_i), \; f_t \in \mathcal{F}, \tag{6}$$

where $(x_i)$ is the training set samples, $f_t$ is the $t$th tress's leaf score, and $\mathcal{F}$ is all classification tress $T$ scores. The results are improved by applying regularization as follows:

$$\mathcal{R}(\sigma) = \sum_i l(y_{hat_i}, \; y_i) + \sum_t \varepsilon(f_t), \tag{7}$$

where $l$ is the cost (loss) function used to calculate the difference between ground truth $y_i$ class labels and the prediction label $y_{hat_i}$. $\varepsilon$ is a function used for penalizing the model's complexity and avoiding overfitting and can be expressed as follows:

$$\varepsilon(f) = \xi K + \frac{1}{2} Y \sum_{j=1}^{K} w_j^2, \tag{8}$$

where $\xi$ and $Y$ are constants for the degree of regularisation, $K$ is the number of leaves on each tree, and $w$ is the leaf weight.

To simplify the objective at step $(t)$, a second-order Taylor expansion is used, as shown below:

$$\mathcal{T}^{(t)} \simeq \sum_{i=1}^{n} [v_i f_i(x_i) + \frac{1}{2} p_i f_i^2(x_i)] + \varepsilon(f)$$

$$= \sum_{i=1}^{n} [v_i f_i(x_i) + \frac{1}{2} p_i f_i^2(x_i)] + \xi K + \frac{1}{2} Y \sum_{j=1}^{K} w_j^2$$

$$= \sum_{j=1}^{K} [(\sum_{i \in I_j} v_i) \, w_j + \frac{1}{2} (\sum_{i \in I_j} p_i + Y) w_j^2] + \xi K, \tag{9}$$

since $I_j = \{i(q(x_i = j)\}$ represents the leaf $t$ instance set,

$$v_i = \frac{\partial l \left( y_{hat_i}{}^{(t-1)}, y_i \right)}{\partial y_{hat_i}{}^{(t-1)}}, \text{ and} \tag{10}$$

$$p_i = \frac{\partial^2 l \left( y_{hat_i}{}^{(t-1)}, y_i \right)}{\partial y_{hat_i}{}^{(t-1)2}}, \tag{11}$$

the optimal weight $w_i$ of leaf $j$ can be calculated for a fixed structure $q(x_i)$ as shown below:

$$w_j = -\frac{\sum_{i \in I_j} v_i}{\sum_{i \in I_j} p_i + Y}, \tag{12}$$

The corresponding optimal value can thus be calculated as follows:

$$\mathcal{T}^{(t)}(q) = -\frac{1}{2} \sum_{j=1}^{K} \frac{\left( \sum_{i \in I_j} v_i \right)^2}{\sum_{i \in I_j} p_i + Y} + \xi K, \tag{13}$$

It is usually difficult to list all of the potential tree architectures *q*, so instead, a greedy method is utilized, which starts with a single leaf and iteratively adds branches to the tree. After the split, let us assume that *IL* and *IR* are the instance sets of left and right nodes. Letting *I* = *IL* ∪ *IR*, the loss reduction after the split can be described as follows:

$$\mathcal{T}_{split} = \frac{1}{2} \left[ \frac{\left( \sum_{i \in IL} v_i \right)^2}{\sum_{i \in IL} p_i + Y} + \frac{\left( \sum_{i \in IR} v_i \right)^2}{\sum_{i \in IR} p_i + Y} + \frac{\left( \sum_{i \in I} v_i \right)^2}{\sum_{i \in I} p_i + Y} \right] - \xi, \tag{14}$$

The XGBoost model's hyperparameters were configured as follows:

Subsample, colsample_bytree, colsam[le_by_level, lambda, and scale_pos_weight are set to 1,

Gamma, and alpha are set to 0, n_estimators: 100, booster: gbtree, max_depth:6, and learning_rate: 0.3.

Furthermore, we noticed that the learning rate increases the training time significantly with little improvement in accuracy when it is less than 0.001. Additionally, the model will be overfitted when increasing the n_estimators and max_depth.

### 3.3. ConvXGB Algorithm

Algorithm 1 describes the complete learning algorithm used in this study.

---

**Algorithm 1:** Training process of the ConvXGB model.

---

**Input:** PTB/MIT-BIH Dataset *D* = {*L*, *y*}
**Output:** The well-trained hybrid neural network *Model*

1:   Train the model using training set $D_{Tr}$;
2:   **for** *start* in range (0, *length* ($D_{Tr}$)) **do**
3:   **for** beat sample $L_i \in D_{Tr}$ **do**
4:   //Multi-lead Attention Module;
5:   $\alpha 1 = ReLU(W_1 L_i + b_1)$;
6:   $X_i = \alpha 1 \otimes L_i$;
7:   //CONVXGB with Attention Mechanism;
8:   $C_1 \leftarrow$ Conv1D($X_i$, *kernels*); *kernel* size: (5, 5) has 16 *kernels* with one stride;
9:   $C_1 \leftarrow$ activation ($C_1$, *ReLU*);
10:  $C_1 \leftarrow$ padding(*same*);
11:  $C_2 \leftarrow$ Conv1D($C_1$, *kernels*); *kernel* size: (5, 5) has 32 *kernels* with one stride;
12:  $C_2 \leftarrow$ activation ($C_2$, *ReLU*);
13:  $C_2 \leftarrow$ padding(*same*);
14:  $C_3 \leftarrow$ Conv1D($C_2$, *kernels*); *kernel* size: (5, 5) has 64 *kernels* with one stride;
15:  $C_3 \leftarrow$ activation ($C_3$, *ReLU*);
16:  $C_3 \leftarrow$ padding(*same*);
17:  M $\leftarrow$ MaxPooling(C3, window); the size of window is (5, 5) with two strides;
18:  XG $\leftarrow$ MaxPooling(M);
19:  $y_{pre} \leftarrow$ XGB Classifier (*XG*);
20:  end for
21:  end for
22:  **return** well-trained *Model*;

---

### 3.4. Performance Measurements

The suggested technique's categorization task is ECG heartbeat categorization for arrhythmia and MI detection. The performance measures used for categorization are accuracy, precision, recall, F1-score, and specificity. These measures are calculated using the following equations:

$$Accuracy = \frac{true_p + true_n}{true_p + true_n + false_p + false_n}, \tag{15}$$

$$Precision = \frac{true_p}{true_p + false_p}, \tag{16}$$

$$Recall = \frac{true_p}{true_p + false_n} \tag{17}$$

$$F1 - score = 2 \times \frac{Precision \times Recall}{Precision + Recall}, \text{ and} \tag{18}$$

$$Specificity = \frac{true_n}{true_n + false_p}, \tag{19}$$

where $true_p$ is the number of instances correctly categorized as required, $false_p$ is the number of cases incorrectly categorized as required, $true_n$ is the number of instances correctly categorized as not required, and $false_n$ is the number of instances incorrectly categorized as not required. Additionally, the AUC–ROC curve is a performance indicator for situations involving categorization with variable threshold values. The AUC value represents the degree or measure of separability, whereas the ROC value represents a probability curve. The AUC–ROC curve represents the model's ability to differentiate across classes. For instance, the higher the AUC is, the more reliably the model predicts zero classes as zero and one class as one. For the MIT-BIH, a multiclass dataset macro averaging technique was used and a simple arithmetic means for all classes' performance metrics. The macro method considers an equal weight for each class.

For the confusion matrix, a cut-off with a threshold value of 50% is used to identify the prediction class. Thus, when the class probability of the prediction is more than 50%, the sample will be classified in that class.

## 4. Experiments

*ECG Datasets*

Tests were carried out using two datasets: the PhysioNet MIT-BIH Arrhythmia dataset [44] and the PTB Diagnostic ECG dataset [45] for heartbeat classification and myocardial infarction classification, respectively. In addition, ECG 2nd lead data resampled at a sampling frequency of 125 Hz were used as an input source. Both datasets are (publicly available) given in Kaggle (https://www.kaggle.com/shayanfazeli/heartbeat, accessed on 16 June 2022), and were utilized in a standardized form and widely used in ECG studies, making it easy to compare results with our proposed method. These datasets have previously been denoised and split into training and testing portions. Furthermore, five classes of arrhythmia and MI localization have already been presented. Figures 3 and 4 show a sample of each class for both datasets. The distribution of training and testing data for each dataset is shown in Table 1.

**Table 1.** Number of training and testing samples in each dataset.

| Dataset | # Samples for Training | # Samples for Testing |
|---|---|---|
| PhysioNet MIT-BIH Arrhythmia dataset | 87,554 | 21,892 |
| PTB | 11,641 | 2911 |

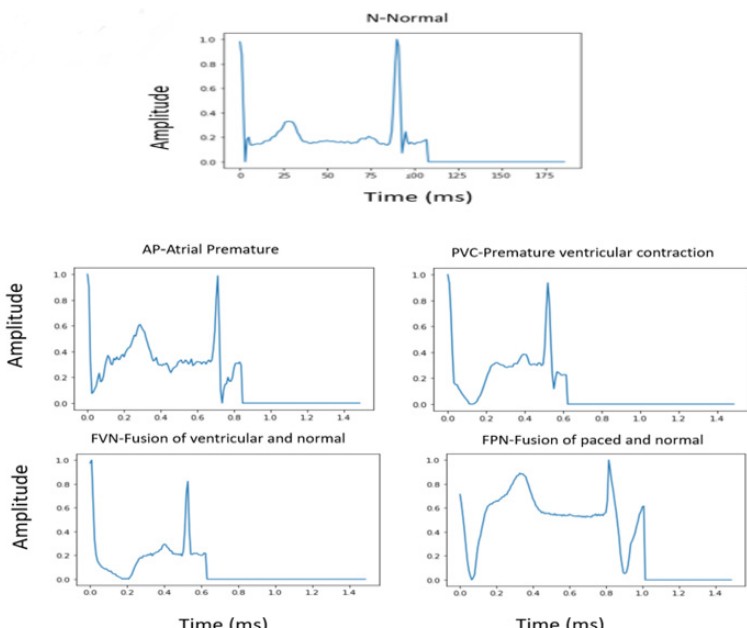

**Figure 3.** Sample plots of five different classes for MIT-BIH dataset.

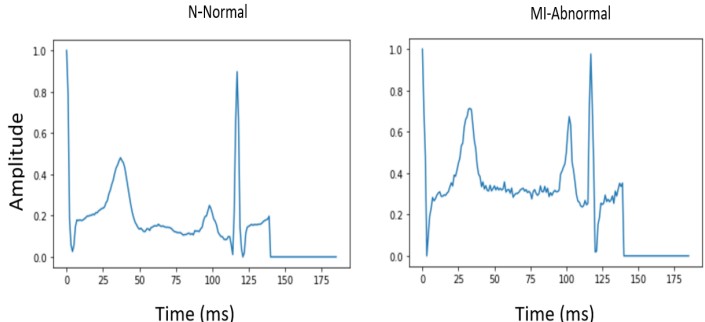

**Figure 4.** Sample plots of heartbeats of two classes for PTB dataset.

The MIT-BIH Arrhythmia dataset is imbalanced, as the distribution of classes shown in Figure 5 illustrates. This affects the classification performance and causes bias towards the class with the highest number of samples. Downsampling of the class with the highest number of samples is used along with upsampling to increase the number of other class samples to avoid overfitting and bias. The class distributions before and after these processes are shown in Table 2.

**Table 2.** Class distributions of MIT-BIH dataset before and after resampling.

| Class Name | # Samples Before | # Samples After |
|---|---|---|
| N | 72,471 | 20,000 |
| AP | 2223 | 20,000 |
| PVC | 5788 | 20,000 |
| FVN | 641 | 20,000 |
| FPN | 6431 | 20,000 |

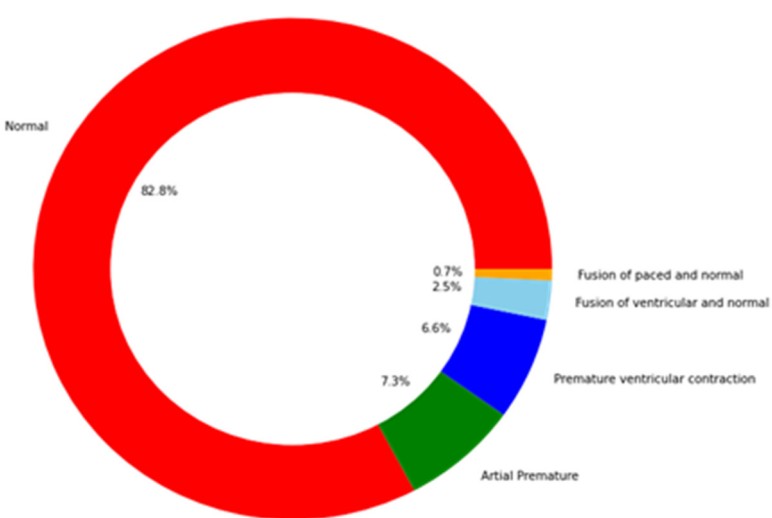

**Figure 5.** MIT-BIH dataset class distribution for the training data.

The description of each class used in MIT-BIH is shown in Table 3.

**Table 3.** Class descriptions for MIT-BIH dataset.

| FPN | FVN | PVC | AP | N |
|---|---|---|---|---|
| Normal | Atrial premature | Premature ventricular contraction | Fusion of ventricular and normal | Paced |
| Left bundle branch block | Aberrant atrial premature | Ventricular escape | | Fusion of paced and normal |
| Right bundle branch block | Nodal (junctional) premature | | | Unclassifiable |
| Atrial escape | Supra-ventricular premature | | | |
| Nodal (junctional) escape | | | | |

The experiments were conducted with an Intel Core i5 processor clocked at 1.7 GHz, 8 GB of RAM, 64-bit Windows 10 Pro, and an NVIDIA GeForce GT 2-GB display card, using Python as the programming language and the TensorFlow library to build the CNN model.

## 5. Results and Discussion

Table 4 summarizes the results of the application of the ConvXGB method to MIT-BIH and PTB datasets in terms of the performance measures previously mentioned.

**Table 4.** Overall performance measurements of ConvXGB.

| Dataset | Accuracy | Precision | Recall | F1-Score | Specificity | Training Time s |
|---|---|---|---|---|---|---|
| MIT-BIH | 0.9836 | 0.9839 | 0.9836 | 0.9837 | 0.9911 | 13.8 |
| PTB | 0.9938 | 0.9938 | 0.9928 | 0.9920 | 0.9948 | 1.23 |

Figure 6 shows the confusion matrices of the proposed model for both datasets. The diagonal entries reflect the percentages of successfully categorized classes, while entries off the diagonal reflect improper categorization. The *x*-axis and *y*-axis represent the predicted labels and actual labels, respectively.

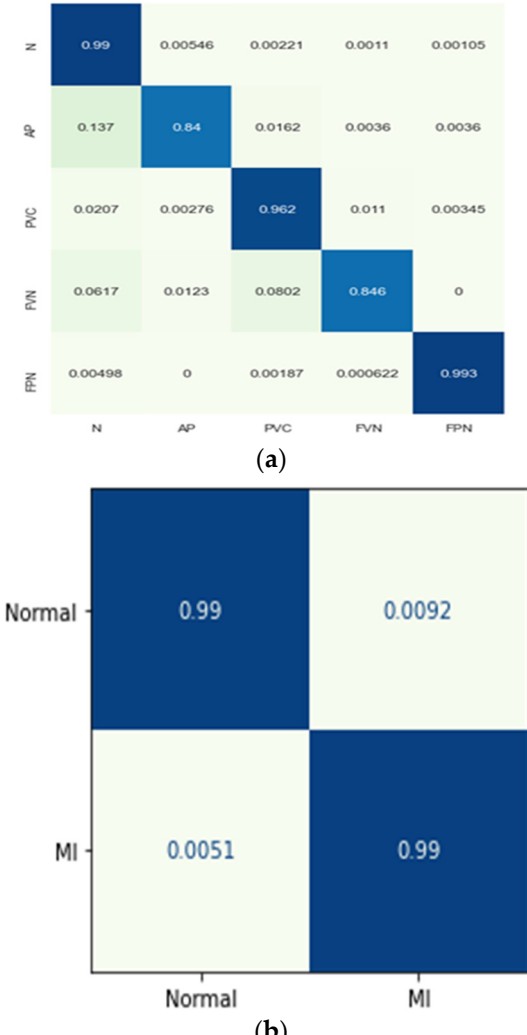

**Figure 6.** Confusion matrices of the ConvXGB model for (**a**) MIT-BIH and (**b**) PTB.

Figure 7 shows the values obtained for AUC, another performance measure used to evaluate the results. Higher values of AUC indicate a better performance at distinguishing between the positive and negative classes. Figure 7 shows that the AUC for the PTB dataset was 100%. The confusion matrix of the model predictions for the PTB dataset shows the low rates of false negatives and false positives (approaching zero), while the result for the MIT dataset indicates some misclassified samples for the AP and FVN classes.

The prediction time for one sample in each dataset was measured and was close for the two datasets (approximately 0.6 ms). This low prediction time indicates that the proposed method is suitable for implementing low-computational-power devices to be developed and applied in the medical field or used as screening tools.

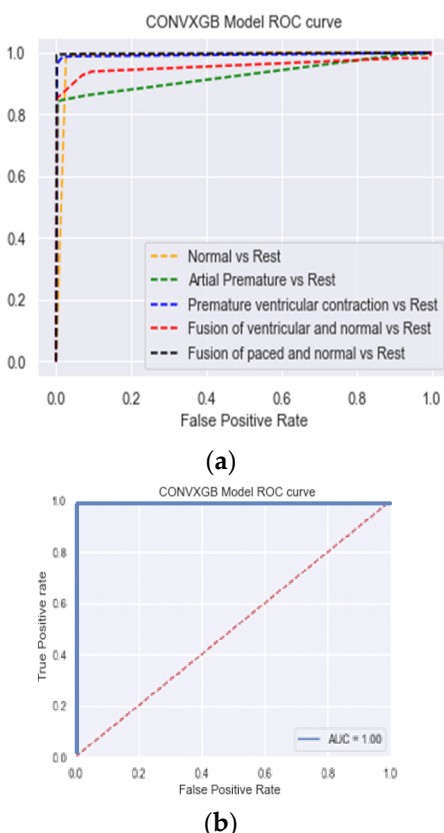

**Figure 7.** AUC of the ConvXGB model for (**a**) MIT-BIH and (**b**) PTB.

## 5.1. Ablation Study

An ablation study was conducted to demonstrate the effectiveness of combining a CNN (for feature extraction) with XGBoost (for classification of the extracted features). We built CNN and XGBoost models separately, with the same architecture as in the ConvXGB model, and the same performance metrics were used. The results are shown in Table 5.

**Table 5.** Performance results for CNN and XGBoost models.

| Dataset | Method | Accuracy | Precision | Recall | F1-Score | Specificity | Training Time s |
|---------|--------|----------|-----------|--------|----------|-------------|-----------------|
| **MIT-BIH** | CNN | 0.9791 | 0.9809 | 0.9791 | 0.9798 | 0.9908 | 490.43 |
| | XGB | 0.9566 | 0.9564 | 0.9567 | 0.9527 | 0.9540 | 131.87 |
| **PTB** | CNN | 0.9948 | 0.9948 | 0.9948 | 0.9948 | 0.9960 | 70.24 |
| | XGB | 0.8980 | 0.9048 | 0.8980 | 0.90 | 0.8906 | 4.03 |

Table 5 shows that the CNN model outperformed the XGBoost model with respect to all performance metrics by at least 2% for the MIT dataset and by approximately 10% for the PTB dataset. The XGBoost model, however, had by far the lowest training time.

On the other hand, the combination of both CNN and XGBoost (our proposed method) produced a better performance than the best performance achieved by the CNN model alone (approximately 1–2% better, as shown in Table 4) and a significantly lower training time than the XGBoost model. These results demonstrate that the proposed ConvXGB method combines the best of both methods and achieves the best performance with the lowest training time.

## 5.2. ConvXGB Comparison with Literature

Table 6 presents the performance results obtained with the most recent state-of-the-art models and the ConvXGB model. The results show that the method proposed in this paper has the highest accuracy, precision, and recall for both datasets used. Some studies have only used accuracy to evaluate model performance, whereas, in this work, several performance measures were used to evaluate the different methods more comprehensively.

**Table 6.** Comparison of the performance of the proposed model with the performance of state-of-the-art models.

| Dataset | Reference | Year | Accuracy | Precision | Recall |
|---------|-----------|------|----------|-----------|--------|
| MIT-BIH | [46] | 2019 | 95.5 | 96.5 | 87.8 |
| | [47] | 2019 | 99.5 | 97.3 | 98.1 |
| | [48] | 2019 | 95.3 | - | - |
| | [49] | 2020 | 96 | - | - |
| | [50] | 2021 | 98.3 | - | - |
| | CNN in this study | 2021 | 0.9791 | 0.9809 | 0.9791 |
| | XGBoostin this study | 2021 | 0.9566 | 0.9564 | 0.9567 |
| | Proposed ConvXGB | 2021 | 0.9836 | 0.9839 | 0.9836 |
| PTB | [51] | 2019 | 83.9 | 82.0 | 95.0 |
| | [51] | 2018 | 96.2 | 97.32 | 93.7 |
| | [52] | 2020 | 96.7 | - | - |
| | [53] | 2020 | 97.7 | - | - |
| | CNN in this studied | 2021 | 0.9948 | 0.9948 | 0.9948 |
| | XGBoostin this study | 2021 | 0.8980 | 0.9048 | 0.8980 |
| | Proposed ConvXGB | 2021 | 0.9938 | 0.9938 | 0.9928 |

It is worth mentioning that for most of the methods compared, high computational power was required, while the proposed method does not require much computational power, and its running time is extremely low. The lower values of recall and precision for most of the state-of-the-art methods indicate higher false-positive and false-negative rates than those of the method proposed in this paper.

The main limitation of the method proposed in this paper is that it is based only on datasets of one-lead ECG signals. We plan to test this method on datasets based on multi-lead ECG signals in the future to overcome this limitation.

## 5.3. Testing of the Method on Another Dataset

To demonstrate the generalizability and robustness of the proposed method, we tested it using another publicly available ECG dataset from the Kaggle website (https://www.kaggle.com/devavratatripathy/ecg-dataset, accessed on 16 June 2022). The dataset contains 2919 normal samples and 2079 MI samples. Each sample represents a complete ECG of a patient with 140 single-lead readings. The results obtained were even better than the results of our experiment, as shown in Table 7.

**Table 7.** The performance measurements of ConvXGB on another dataset.

| Accuracy | Precision | Recall | F1-Score | Specificity |
|----------|-----------|--------|----------|-------------|
| 0.994 | 0.9935 | 0.9937 | 0.9936 | 0.9933 |

The confusion matrix in Figure 8 shows the low rates of false positives and false negatives (approaching zero), and Figure 9 shows the ROC curve, with the area under the curve (AUC) approaching 100%.

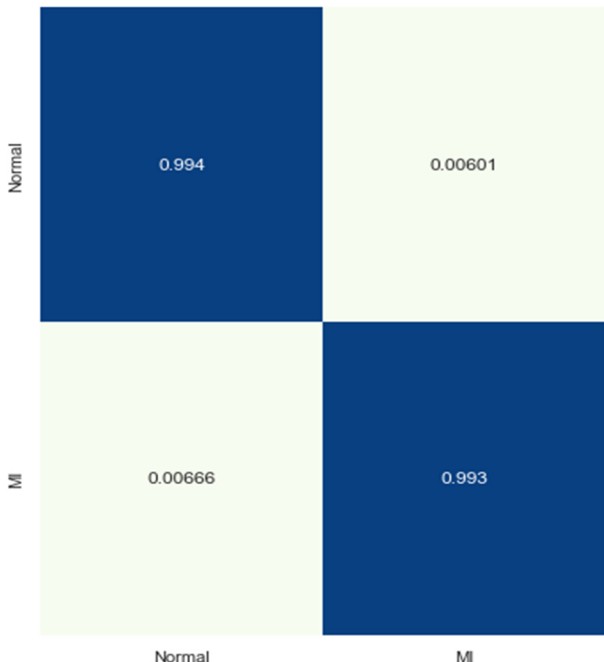

**Figure 8.** Confusion matrix of the ConvXGB model for another dataset.

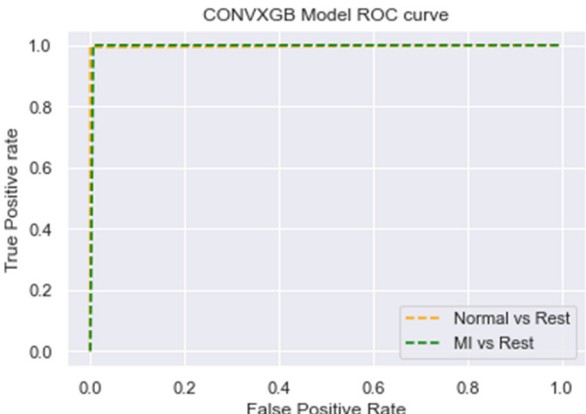

**Figure 9.** AUC of the ConvXGB model for another dataset.

## 6. Conclusions

The accurate classification of ECG waves is exceptionally beneficial in preventing and detecting cardiovascular diseases. By integrating medical and contemporary machine learning technologies, deep CNNs have proven to be highly effective in improving the accuracy of cardiovascular disease diagnosis through ECG signal feature extraction. Similarly, the XGBoost algorithm has demonstrated its exceptional ability in classification. We propose a new model, ConvXGB, that achieves both high computational efficiency and high accuracy by combining the CNN and XGBoost methods. The results of experiments conducted using PhysioNet's MIT-BIH dataset for five distinct arrhythmias and the PTB diagnostics dataset for MI classification show that the proposed hybrid model is superior to both of its component models. The proposed model also outperforms existing state-of-the-art classification methods in terms of accuracy, precision, and recall. The most notable finding of the study is that using ConvXGB improves machine learning task performance compared to either approach used separately. Our proposed method correctly classified arrhythmias 99.38% of the time. This result demonstrates that the proposed ConvXGB approach is highly successful in classifying arrhythmia. As a limitation, we recommend re-training the

model when using real-world data since the experiments conducted in the ideal datasets are not comparable to data from the real world.

In future research, this method should be fine-tuned and modified for use in real-time systems to classify heartbeat signals to advise medical experts. In addition, it would be more efficient to use multi-channel signals rather than depending on just one lead's signal.

**Author Contributions:** Conceptualization, A.A.R., M.K.E. and A.M.A.; methodology, A.A.R., M.K.E. and A.M.A.; software, A.A.R., M.K.E. and A.M.A.; validation, A.A.R., M.K.E. and A.M.A.; formal analysis, A.A.R., M.K.E. and A.M.A.; investigation, A.A.R., M.K.E. and A.M.A.; resources, A.A.R., M.K.E. and A.M.A.; data curation, A.A.R., M.K.E. and A.M.A.; Writing and original draft preparation, A.A.R., M.K.E. and A.M.A.; writing—review and editing, A.A.R., M.K.E. and A.M.A.; visualization, A.A.R., M.K.E. and A.M.A.; supervision, A.A.R., M.K.E. and A.M.A.; project administration, A.A.R., M.K.E. and A.M.A.; funding acquisition, A.A.R., M.K.E. and A.M.A. All authors have read and agreed to the published version of the manuscript.

**Funding:** This research received no external funding.

**Acknowledgments:** The authors would like to acknowledge the University of Gezira for the support it provides, and also, we would like to thank all who gave us support to complete this paper.

**Conflicts of Interest:** The authors declare no conflict of interest.

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
