# Peer review of "ECG Heartbeat Classification Using CONVXGB Model"

_electronics, doi:10.3390/electronics11152280_

Round 1

Reviewer 1 Report

Report on “ECG Heartbeats Classification Using CONVXGB Model” by Atiaf A. Rawiet al.

The manuscript does a modeling exercise, using preprocessed databases, through the use of convolutional networks and additive models based on classification trees.

The introduction is extensive and the material and methods section introduces in an orderly and clear way both the models used and the classification parameters.

My main concern lies in the reproducibility of the results, since the validation parameters provide values that are too high and difficult to reproduce in real data that require filtering and preprocessing of the signals.

Another concern is the lack of information about the XGBoost modeling architecture. In the figure 2 and the code provided we can see how the convolutional type network has been built, but in the XGBoost algorithm, what is the number of trees used in the summation, maximum depth of the tree, sample size or the number of features used in each tree. Please complete. Also, It must be considered to analyze the variable importance in the XGBoost model.

It also has to be explained how the validation parameters for MIT-BIH have been calculated, because this is not a binary classification problem. If it has been calculated as the average of the binary prediction of the 5 classes, this has to be clarified in material and methods.

In the case of the confusion matrix, it has not been explained what the cut-off point that has been taken to carry out the classification is, is it the Youden index? A certain probability? please clarify

Finally, I recommend adding as limitation explaining that results must be take with caution as no application on real data was made, thus results must be overoptimistic.

Author Response

Dear Sir

All the notes were done, please find the attached file

Thanks and Regards

Reviewer 2 Report

Why MIT-BIH and PTB are chosen and have you tried the CONVXGB Model using other datasets, preferable fresh ECG, which are not taken from the online dataset. Some spelling errors, combined words, are they not checked before submission?

Author Response

Dear Sir

All the comments were done, please find the attached file

Thanks and Regards

Round 2

Reviewer 1 Report

All my concerns have been addressed by the authors.